# Correlation among Routinary Physical Activity, Salivary Cortisol, and Chronic Neck Pain Severity in Office Workers: A Cross-Sectional Study

**DOI:** 10.3390/biomedicines10102637

**Published:** 2022-10-19

**Authors:** Juan Antonio Valera-Calero, Umut Varol

**Affiliations:** 1Department of Physiotherapy, Faculty of Health, Camilo Jose Cela University, 28962 Villanueva de la Cañada, Spain; 2VALTRADOFI Research Group, Department of Physiotherapy, Faculty of Health, Camilo Jose Cela University, 28962 Villanueva de la Cañada, Spain; 3Escuela Internacional de Doctorado, Universidad Rey Juan Carlos, 28922 Alcorcón, Spain

**Keywords:** hydrocortisone, neck pain, chronic pain, sedentary behavior, exercise

## Abstract

This study aimed to assess the correlation between different aspects of routinary physical activity with chronic neck pain severity indicators and salivary cortisol. This cross-sectional observational study included ninety-four office workers with non-specific chronic neck pain in the analyses. Pain related outcomes (pain intensity, pressure pain thresholds and disability), physical activity outcomes using the International Physical Activity Questionnaire, and salivary cortisol levels were evaluated. Pearson’s correlation analysis was used to investigate internal associations and regression models to explain and calculate which factors contribute to the variance of salivary cortisol and neck pain severity. Female sex (*p* < 0.01), sedentary behaviors (*p* < 0.05), and pain sensitivity (*p* < 0.05) were associated with greater cortisol levels (*p* < 0.05), but disability and pain intensity were not associated (*p* > 0.05). Worse disability, pain intensity, and pain pressure thresholds were also associated with lower routinary physical activity (*p* < 0.05). Regression models explained 20.6% of pain intensity (based on walking time to their workplace, age and pain sensitivity); 27.3% of disability (based on moderate physical activity at home, vigorous physical activity during leisure time and pain sensitivity); 54.2% of pain sensitivity (based on cycling time from home to their workplace, gender and vigorous activity during leisure time) and 38.2% of salivary cortisol concentration (based on systolic pressure, vigorous activity at work and both moderate and vigorous activity at home). Our results demonstrated the association between salivary cortisol concentration with moderate and vigorous physical activity, sitting time at work, and PPTs. However, salivary cortisol was not associated with disability or pain intensity.

## 1. Introduction

Neck pain is defined as the pain perceived in the area delimited on top by the external occipital protuberance, below with an imaginary transverse line across T1 spinous process, and laterally with vertical lines tangential to the lateral borders of the neck [1,2]. It has a major physical, psychological, and socioeconomic impact, as it is the fourth most frequent cause of disability, preceded of low back pain, depression, and arthralgia [3]. In fact, up to 50–70% of the entire population will experience (at least) one episode of neck pain clinically important throughout their life [2].

Assuming normal conditions, cortisol secretion occurs during an acute stress episode as a result of fight or flight response that also facilitates the formation of a memory related to fear for facilitating survival [4]. In addition to these secretion peaks, as a consequence of a stress episode, there are physiological secretions modulated by sleep/wake circadian rhythms [5]. In both cases, cortisol is the hormonal endpoint of the hypothalamic-pituitary-adrenal axis (HPA-Axis), a complex regulatory mechanism that utilizes feed-forward and feed-back loops, and interacts with the sympathetic autonomic nervous system and the immune system to regulate circadian metabolic, cognitive system, cardiovascular, and immunological responses to maintain an adequate homeostasis [6].

Several prospective cohorts’ research reported that high levels of anxiety are significant predictors of pain, depression, and lower life quality [7], while pain is a factor that could trigger the stress response by itself [8]. Prolonged or recurring stress exposure, pain catastrophism, and fear-avoidance behaviors can trigger variable responses to pain thresholds’ intolerance depending on the magnitude of the individual stress response [9], and therefore elevate cortisol secretion [10]. This maladaptive hypervigilance response to stressful stimuli results in prolonged activations of cortisol secretions [10,11].

On the other hand, current evidence assessing the relationship between physical activity and pain is clear and consistent. Those subjects who report higher levels of vigorous or total physical activity are more likely to reduce or prevent general chronic pain [12,13]. Even populations with high central sensitization such as fibromyalgia evading sedentary behaviors report less pain [14,15]. In addition, subjects with chronic low back pain showing fear-avoidance behaviors and depression are more likely to suffer more severe disability [16]. Although jobs with greater physical loads do not affect the risk or prognosis of neck pain [17], physical activity during leisure time has a protective effect on the risk of chronic neck pain [18,19,20]. Regular physical activity is associated with greater pain pressure thresholds and better general mechanosensitivity and pain tolerance in specific neck locations, preventing the onset of chronic pain [18,19,20]. However, physical activity levels have no effect on the acute pain onset [20].

Finally, previous research analyzed the immediate effect of exercise on salivary [21,22,23] and hair [24] cortisol concentration. For instance, after yoga performance, there is a significant immediate decrease in the cortisol secretion [21], which is positively associated with fatigue and being overweight [22,23,24]. Regarding the medium term, previous research reported adaptations in cortisol secretion after an 8-week exercise routine [25]. However, despite lumbar stabilization, exercises have a direct effect on β-endorphins levels, and no effect on the cortisol secretion was found [26]. A review collected data on several studies considering different types of exercise and different samples, but the inconsistency of data makes it necessary to assess more homogeneous and consistent samples, controlling the exercise loads [27,28]. 

Since evidence is controversial and due to the lack of studies assessing the correlation between physical activity, salivary cortisol, and chronic neck pain severity in stressful jobs such as office workers, the objectives of this research were to assess whether different aspects of physical activity, pain indicators, and salivary cortisol concentration are internally correlated and to calculate a linear regression model for identifying those attributes explaining the salivary cortisol and neck pain severity variance.

## 2. Materials and Methods

### 2.1. Study Design

An observational cross-sectional study research was conducted at a private clinic located in Madrid (Spain) to calculate the association between sociodemographic data, chronic neck pain, salivary cortisol, and physical activity. This report followed the Strengthening the Reporting of Observational studies in Epidemiology (STROBE) guidelines and checklist [29]. In addition, we considered all the recommendations stated in the Declaration of Helsinki and the Clinical Ethical Committee of Universidad de Alcalá. All participants signed a written informed consent prior to the start of the study. 

### 2.2. Participants

A consecutive sample of office workers was recruited and screened for potential eligibility between April 2021 and September 2021. To be included in the study, participants had to be between 18 and 65 years old, and suffer continuous neck pain during at least 6 months within the last year with a minimum intensity of 3/10 points in the Visual Analogue Scale and at least 8% of disability in the Neck Disability Index. Participants were excluded if they presented any of the following conditions: (1) Receiving hormonal therapy (cortisol responses to stress and circadian regulations showed to be altered during contraceptive hormonal intake [30]); (2) Presenting any pain sensitivity alteration or condition; (3) Suprarenal gland disturbances; (4) Overage or deficiency of basal cortisol due to pathological conditions; and (5) pregnancy.

### 2.3. Sample Size Calculation

Considering this study as a prognostic study, a range from 10 to 15 participants per potential predictor (with no more than five predictor variables for avoiding overestimation of the results), a minimum sample size of 75 participants could be considered appropriate [31]. Due to the cross-sectional nature of this study, no losses were considered.

### 2.4. Outcomes

#### 2.4.1. Demographic and Vital Signs 

Participants filled out a self-reported document containing age, gender, height, and weight information. Later, the body mass index (BMI) was calculated for each participant. Data related to vital signs were collected using a pulse oximeter NANOXμ^®^ for oxygen saturation and an OMRON M3^®^ device for heart rate and blood pressure.

#### 2.4.2. Salivary Cortisol

Measuring cortisol as a stress biomarker in saliva demonstrated to be a valid and convenient alternative to plasma cortisol measurement [32] reflecting the HPA axis activity [33,34]. In addition, this method has several advantages in comparison with plasma measurements since samples can be collected during normal daily routines, stress-induced cortisol release responses are less likely to occur, and fewer human resources are needed [35]. In addition, previous studies demonstrated its reliability [36].

Since salivary concentration is regulated by circadian rhythms, all sample collections were programmed at the same time (between 10:00 a.m. and 10:30 a.m.) for avoiding timing bias. Patients were asked not to eat, drink, chew gum, or brush their teeth 60 min prior to the sample collection and asked to rinse their mouth with cold water 5 min prior to the sample collection. A minimum of 1.0 mL of saliva was required for processing the sample. Samples with a reddish color were dismissed due to possible blood contamination [33,34,35,36]. 

The instrument used for analyzing the saliva samples was a Cortisol kit RE52611 ELISA^®^. All saliva samples were stored at −20 °C until analysis. To analyze the cortisol level in the saliva, four steps were followed. First, 50 μL of saliva was added to a tube using a pipette, and 100 μL of enzyme conjugate was added to each tube. Then, closed tubes were incubated at room temperature (18–25 °C) for 2 h in an orbital shaker (orbital shaker S-3.02 10L, ELMI SIA, Riga, Latvia) at 400–600 rpm. Adhesive foils were removed, and the incubation solution was discarded. Plates were washed with 250 μL of diluted wash buffer, and the excess solution was removed by tapping the inverted plate on a paper towel. Then, 100 μL of TMB substrate solution was pipetted and incubated 30 min at room temperature on an orbital shaker (again, 400–600 rpm). Afterward, 100 μL of TMB inhibitory solution was added (samples turned from blue to yellow), and results were measured using the EASIA microplate reader (Medgneix, Fleurus, Belgium) at 450 nm reading absorbance within 15 min after adding the last reagent. 

#### 2.4.3. Chronic Neck Pain Severity

Neck pain severity was analyzed using the Neck Disability Index (NDI) for assessing the neck disability as is recognized as a valid and high-reliable test [37], the Visual Analogue Scale (VAS) performing a mean average of three measures (pain during the exam, worst and best pain during the previous week) for assessing the pain intensity [38], and pressure pain thresholds (PPT) using an electronic algometer (Wagner Force Dial^®^ FDK/FDN Series) over the most symptomatic zygapophyseal joint performing a mean average of 3 different trials for assessing pain sensitivity [39].

#### 2.4.4. Physical Activity

Physical activity was measured using the International Physical Activity Questionnaires (IPAQ), assessing the last 7 days of physical activity [40]. This is a validated questionnaire for assessing the time of physical activity during working, at home, in leisure time. It also assesses sitting time during working and during leisure time.

### 2.5. Statistical Analysis

All statistical analyses were conducted in IBM SPSS Statistics Version 27 for Mac OS (IBM Corporation, Armonk, NY) setting a significance level of *p* < 0.05. After verifying data distribution using the Shapiro–Wilk test, descriptive statistics were used to characterize the sample. Normal-distributed data were described by mean and standard deviation (SD) and non-normal distributed data were descriptively presented as median and interquartile range. 

Pearson’s Correlation Coefficients (*r*) were calculated to estimate the bivariate correlations and to identify multicollinearity and shared variance (*r* > 0.80) between the variables. One matrix was structured to identify the correlation between salivary cortisol with routinary physical activity, one between salivary cortisol with demographic and clinical characteristics, and one between physical activity with pain-related outcomes.

Finally, four linear regression models were calculated for explaining the variance of pain intensity, disability, pain sensitivity, and salivary cortisol concentrations. Each model was built including in a stepwise multiple regression model (hierarchical regression analysis) all the variables contributing significantly to the variance of each dependent variable. To be considered as a potential predictor, *F* values should have a significance level *p* < 0.05. Changes in adjusted R^2^ were reported step by step to determine the individual contribution of each predictor.

## 3. Results

From 551 patients recorded in the Campus Repsol Physical Therapy Department’s database, 154 office workers were potentially eligible to be identified by chronic pain complaints. After checking their self-reported pain intensity and disability, 96 participants were confirmed for eligibility and included in the study. Due to unknown pregnancy during the evaluation, two participants were excluded of the statistical analysis. Therefore, a total of 94 participants (20 males and 74 females) were finally included in the analyses. 

Demographic, clinical, and physical activity data obtained from the total sample and divided by gender are summarized in Table 1. In general, the sample had mild neck disability, moderate pain intensity, and physiological blood pressure and heart rate at rest. Although men exhibited greater weight (*p* < 0.001), height (*p* < 0.001), BMI (*p* < 0.05), and PPT (*p* < 0.001) compared with women, females were older (*p* < 0.05) and showed a higher heart rate at rest (*p* < 0.001) than males.

Correlations between salivary cortisol with routinary physical activity levels are reported in Table 2. Results demonstrated significant negative associations between cortisol with vigorous and moderate activity at the workplace (*p* < 0.05), vigorous and moderate activity at home (*p* < 0.05), and vigorous and moderate activity in leisure time (*p* < 0.01). In contrast, a larger sitting time at workplace was associated with greater salivary cortisol levels (*p* < 0.01).

Pearson’s correlation coefficients between cortisol, demographic, and clinical characteristics are described in Table 3. Greater cortisol concentration was positively correlated with the female sex and a higher heart rate at rest (both, *p* < 0.01). In addition, salivary cortisol was negatively correlated with weight (*p* < 0.01), BMI (*p*< 0.01), systolic pressure (*p* < 0.01), and PPT (*p* < 0.05). Despite pain sensitivity being associated with pain intensity (*p* < 0.05) and disability (*p* < 0.01), pain intensity and disability were not significantly associated (*p* > 0.05).

Table 4 describes Pearson’s correlation coefficients between physical activity features and neck pain severity outcomes. NDI showed a positive correlation with moderate physical activity at home (*p* < 0.05) and a negative correlation with vigorous physical activity during leisure time (*p* < 0.01); VAS was positively correlated with sitting time at work (*p* < 0.05) and negatively correlated with both vigorous (*p* < 0.05) and moderate (*p* < 0.01) physical activity during leisure time and walking time from home to their workplace (*p* < 0.05); PPT was positively correlated with vigorous physical activity at home (*p* < 0.01), work and leisure time (also moderate) and cycling from home to their workplace (all, *p* < 0.05).

In addition, significant correlations also existed among the independent physical activity features, demographic, and clinical characteristics with no multicollinearity (*r* > 0.80 except for BMI and weight). Table 5 shows the hierarchical regression analysis conducted in this study for pain outcomes. For VAS, walking time to their working place contributed 3.4% of the variance (*p* = 0.008), age contributed an additional 7.2% (*p* = 0.002), and PPT the last 10.0% of variance (*p* = 0.002). When combined, physical activity, demographic, and clinical features explained 20.6% of the VAS in chronic pain patients (*p* < 0.001). 

For NDI, moderate physical activity at home contributed 5.5% of variance (*p* = 0.008), vigorous activity during leisure time contributed an additional 9.5% of the variance (*p* = 0.024) and PPT the last 12.3% (*p* < 0.001). When combined, physical activity and clinical features explained 27.3% of the variance of NDI (Table 5).

For PPT, NDI contributed 16.9% of variance (*p* < 0.001), cycling time from home to their workplace contributed an additional 4.1% of the variance (*p* < 0.001), gender a 25.0% (*p* < 0.001), and vigorous physical activity at home the last 6.4% (*p* = 0.001). When combined, physical activity, demographic, and clinical features explained 54.2% of the variance of PPT (Table 5).

Table 6 shows the hierarchical regression analysis to determine salivary cortisol levels. Systolic pressure contributed 18.4% of the variance (*p* < 0.001), vigorous activity at work contributed an additional 8.1% (*p* = 0.008), moderate physical activity an additional 8.1% (*p* < 0.001), and vigorous physical activity at home the last 3.6% (*p* = 0.022). When combined, clinical and physical activity features explained 38.2% of the variance of the salivary cortisol level in patients with chronic neck pain.

## 4. Discussion

### 4.1. Cortisol and Pain

Although a previous report demonstrated a tendency towards higher cortisol levels in patients with neck pain compared with asymptomatic subjects, widespread chronic pain (e.g., fibromyalgia syndrome) showed lower levels than healthy controls [41], and the association between cortisol levels and chronic pain conditions remains unclear [42]. 

Our results showed whether subjects with lower basal cortisol levels have more pressure pain tolerance. On the other hand, data showed no associations between pain intensity or disability with basal cortisol levels. The inconsistency between this study and the current evidence might be due to cortisol secretions produced during acute pain episodes, due to the baseline disability indexes of the samples or higher subjective pain perceptions to produce a maladaptive function [9,10,11] or an absence of depression, anxiety, bad life quality, or fear-avoidance to pain (factors associated with higher cortisol basal levels).

Furthermore, depression, fear-avoidance to pain, catastrophic behaviors, anxiety, and life quality were not assessed in this research, being outcomes associated with increased basal cortisol levels [12]. Therefore, a good pain tolerance and/or absence of depression, anxiety or bad life quality could explain an absence of a maladaptive cortisol response.

Although cortisol levels were different in regional (hypercortisolism) and widespread (hypocortisolism) musculoskeletal pain [41], depression and stress-related conditions are closely associated with greater cortisol levels [43,44]. 

Finally, many clinical trials assessed whether different interventions induce changes in biomarkers and pain severity [45,46,47,48,49]. For example, massage is a technique supported by evidence to reduce cortisol, depression, pain syndromes, stress conditions, and immune chronic illnesses [45]. In addition, joint manipulations and mobilizations can induce salivary cortisol changes [46,47,48,49]. However, the salivary cortisol effects are different depending on the population (i.e., in asymptomatic subjects, there is a decrease in cortisol concentration 6 h after the intervention [46] while mobilization and manipulation induce a comparable immediate increase in salivary cortisol [47] in patients with chronic neck pain), segments targeted (i.e., plasma cortisol increase after cervical manipulation, but not after thoracic manipulation [48]) and patients’ expectations and beliefs (i.e., salivary cortisol increased in negative and neutral expectations in contrast with positive expectations [49]).

### 4.2. Physical Activity and Pain

Current evidence related to physical activity and pain is clear. Those subjects who self-reported higher levels of vigorous or total physical activity are more predisposed to reduce or prevent chronic general pain [12,13,18,19], and those subjects with lower sedentary attitudes present also lower pain levels [14,15,16]. However, since pain is a complex experience modulated by multiple biopsychosocial factors [50], an independent evaluation of PPTs, disability, and pain intensity factors are needed [51].

Regarding the association between routinary physical activity and disability, Lee et al. [52] analyzed the association between the IPAQ and neck disability (assessed with the Northwick Park Neck Pain Questionnaire) and found no significant associations. Since they did not differentiate physical activity type, this may explain the results’ differences with our study. In fact, Kim et al. [53], in accordance with our results, also found physical activity during leisure time to be a protective factor for neck disability.

On the other hand, a previous meta-analysis describing the association between physical activity and pain intensity [54] found no association between physical activity during leisure time and neck pain but walking or cycling for at least 150 min/week might have a favorable effect on neck symptoms. However, these conclusions were made based on a single high-quality study. Additionally, one observational study declared that physical activity at work does not affect the risk or prognosis of neck pain [17]. In contrast with these studies, our results showed whether those subjects with sedentary behaviors reported worse neck pain intensity. Physical therapists should include a physical activity program to reduce the subjective perception of pain and increase PPT in office workers with chronic neck pain.

Finally, a recent systematic review demonstrated that exercise modulates neck and remote PPTs in patients with shoulder pain [55]. Similarly, we found lower subjective pain perception and higher PPT in those subjects who perform more vigorous or moderate physical activity at work and at home.

### 4.3. Physical Activity and Cortisol

Previous evidence is not consistently related to the relationship between physical activity and cortisol [27,28]. Most of the studies were performed to assess whether cortisol secretion changes during and after physical activity, and evidence related to adjustments in basal cortisol levels depending on the physical activity routine is limited.

Cortisol secretion seems to oscillate depending on the time, intensity, specificity, and emotional/psychological effects of the physical activity, where moderate-to-high intensity exercise produces a secretion response and low-intensity produces no responses [56,57]. For instance, while a one-hour session of power or stretch yoga reduces the salivary cortisol concentration similarly (although power yoga was perceived to be more pleasurable and energizing in comparison with stretch yoga) [21], three tested core stabilization exercises (crook lying and rest, passive cycling in crook lying using an automatic cycler, and a lumbar core stabilization exercise on a Pilates device) produced no changes in plasma cortisol concentration [26]. Our results showed lower basal cortisol levels in those subjects who performed vigorous and/or moderate physical activity at home, work, and leisure time and a greater cortisol concentration in those subjects with sedentary behaviors. A physical activity routine is recommended to decrease cortisol level, stress, and anxiety. Further studies should assess optimal loads of exercise to modulate the inhibition of cortisol secretion with different samples.

Being overweight and fatigue, which are closely associated with poor physical activity [58], are also reported to be positively associated with cortisol concentrations [22,23,24]. Since our ranges of body mass index and weight were limited, this may explain the inverse association found with salivary cortisol and therefore the associations found should be carefully interpreted. 

## 5. Conclusions

These findings indicate that those office workers with chronic neck pain and mild disability who presented lower cortisol levels showed higher PPTs, but no higher or lower subjective pain perception or disability. Furthermore, those patients who performed more vigorous and moderate physical activity as routine presented lower cortisol levels, pain intensity, and higher PPTs. Those participants who performed more moderate activity at home presented more disability.

## Figures and Tables

**Table 1 biomedicines-10-02637-t001:** Baseline characteristics of the participants.

Variables	Sample	Males	Females
Demographic characteristics			
Smoking, n (yes (cigarettes) /no)	22 (6.4 ± 4.9)/72	8 (6.3 ± 4.9)/12	14 (6.4 ± 4.9)/60
Weight (kg) *	66.6 ± 10.1	77.3 ± 6.5	63.7 ± 8.8
Height (m) *	1.67 ± 0.07	1.74 ± 0.04	1.64 ± 0.06
BMI (kg/m^2^) **	23.9 ± 3.4	25.3 ± 2.6	23.6 ± 3.5
Age (years) **	38.1 ± 8.7	34.6 ± 5.8	39.1 ± 9.1
Clinical characteristics			
NDI (0–100)	24.9 ± 10.0	25.2 ± 8.9	24.8 ± 10.3
VAS (0–10)	6.1 ± 1.9	5.7 ± 1.9	6.3 ± 1.9
PPT (kPa) *	193.2 ± 71.8	271.7 ± 70.3	172.0 ± 56.0
Physiological characteristics			
Systolic pressure (mm Hg)	119.5 ± 10.8	120.2 ± 11.1	118.9 ± 10.9
Diastolic pressure (mm Hg)	73.9 ± 8.8	74.6 ± 11.4	73.2 ± 8.0
Heart rate at rest (bpm) *	65.0 ± 9.1	59.0 ± 7.8	66.5 ± 8.8
Cortisol (µg/dL)	0.79 ± 0.57	0.69 ± 0.61	0.81 ± 0.55

Abbreviations: BMI: Body Mass Index; NDI: Neck Disability Index; PPT: Pain Pressure Threshold; VAS: Visual Analogic Scale. * *p* < 0.001; ** *p* < 0.05.

**Table 2 biomedicines-10-02637-t002:** Pearson-product moment correlation matrix: cortisol and physical activity.

	1	2	3	4	5	6	7	8	9	10	11	12	13
**1. Cortisol**													
**2. Vigorous at workplace**	−0.273 *												
**3. Moderate at workplace**	−0.277 *	n.s											
**4. Walking at workplace**	n.s	0.350 **	0.419 **										
**5. Car: home to workplace**	n.s	n.s	n.s	n.s									
**6. Cycling: home to workplace**	n.s	n.s	n.s	n.s	n.s								
**7. Walking: home to workplace**	n.s	0.578 **	n.s	0.350 **	n.s	n.s							
**8. Vigorous at home**	−0.231 *	0.896 **	n.s	0.292 **	n.s	n.s	0.508 **						
**9. Moderate at home**	−0.265 *	n.s	0.343 **	n.s	n.s	n.s	0.407 **	n.s					
**10. Walking in leisure time**	n.s	n.s	n.s	n.s	−0.247 *	n.s	n.s	n.s	n.s				
**11. Vigorous in leisure time**	−0.298 **	n.s	n.s	0.288 **	0.412 **	n.s	n.s	n.s	n.s	n.s			
**12. Moderate in leisure time**	−0.326 **	0.474 **	n.s	n.s	n.s	n.s	0.419 **	0.479 **	n.s	n.s	0.379 **		
**13. Sitting at workplace**	0.315 **	n.s	n.s	n.s	n.s	n.s	n.s	n.s	n.s	n.s	−0.373 **	−0.413 **	
**14. Sitting in leisure time**	n.s	n.s	n.s	n.s	0.503 **	−0.248 *	n.s	n.s	n.s	−0.335 **	n.s.	n.s.	0.305 **

Note: 1–14 are the same as the numbers/item of the Y aches; values are Pearson’s r score if significant associations were found. n.s. non-significant, * *p* < 0.05; ** *p* < 0.01.

**Table 3 biomedicines-10-02637-t003:** Pearson-product moment correlation matrix: cortisol, demographic, clinical and physiological characteristics.

		1	2	3	4	5	6	7	8	9	10	11	12
	1. Cortisol												
**Demographic** **Characteristics**	2. Gender	0.321 **											
	3. Weight	−0.303 **	−0.510 **										
	4. Height	n.s	−0.562 **	0.443 **									
	5. Age	n.s	0.339 *	n.s	n.s								
	6. BMI	−0.333 **	n.s	0.823 **	n.s	n.s							
	7. Smoking	n.s.	n.s.	n.s.	n.s.	n.s.	n.s.						
**Physiological** **Characteristics**	8. Systolic pressure	−0.441 **	n.s	0.402 **	n.s	n.s	0.422 **	n.s.					
	9. Diastolic pressure	n.s	n.s	n.s	n.s	n.s	0.229 *	n.s.	0.536 **				
	10. Heart rate	0.284 **	0.416 **	n.s	−0.295 **	n.s	n.s	n.s	n.s	n.s			
**Clinical** **Characteristics**	11. NDI	n.s	n.s	n.s	−0.262 *	n.s	n.s	n.s	n.s	n.s	n.s		
	12.VAS	n.s	n.s	n.s	−0.203 *	−0.275 *	n.s	n.s	n.s	n.s	n.s	n.s	
	13. PPT	−0.259 *	−0.545 **	0.526 **	0.320 **	n.s	0.406 **	n.s	n.s	n.s	n.s	−0.424 **	−0.267 *

Note: 1–13 are the same as the numbers/item of the Y aches; values are Pearson’s r score if significant associations were found. Abbreviations: BMI: Body Mass Index; NDI: Neck Disability Index; PPT: Pain Pressure Threshold; VAS: Visual Analogic Scale n.s. non-significant, * *p* < 0.05; ** *p* < 0.01.

**Table 4 biomedicines-10-02637-t004:** Physical activity associations with pain-related outcomes.

Physical Activity Variables	NDI	PPT	VAS
Vigorous at workplace	n.s	0.276 *	n.s
Moderate at workplace	n.s	n.s	n.s
Walking at workplace	n.s	n.s	n.s
Car from home to workplace	n.s	n.s	n.s
Cycling from home to workplace	n.s	0.252 *	n.s
Walking from home to workplace	n.s	n.s	−0.234 *
Vigorous at home	n.s	0.348 **	n.s
Moderate at home	0.258 *	n.s	n.s
Walking in leisure time	n.s	n.s	n.s
Vigorous in leisure time	−0.320 **	0.254 *	−0.222 *
Moderate in leisure time	n.s	0.220 *	−0.245 **
Sitting at workplace	n.s	n.s	0.220 *
Sitting in leisure time	n.s	n.s	n.s

Abbreviatures: NDI: Neck Disability Index; PPT: Pain Pressure Threshold; VAS: Visual Analogic Scale. n.s. non-significant. * *p* < 0.05; ** *p* < 0.01.

**Table 5 biomedicines-10-02637-t005:** Summary of the regression analyses to determine pain outcomes.

Targeted Variable	Predictor Outcome	*B*	SE B	95% CI	*β*	*t*	*p*-Value
VAS	Step 1 *Walking to workplace*	−0.216	0.004	−0.017, 0.000	−0.008	−1.949	0.05
	Step 2 *Walking to workplace* *Age*	−0.230 −0.287	0.004 0.017	−0.017, −0.001 −0.080, −0.012	−0.009 −0.046	−2.163 −2.690	0.034 0.009
	Step 3 *Walking to workplace* *Age* *PPT*	−0.278 −0.325 −0.333	0.004 0.016 0.010	−0.019, −0.003 −0.084, −0.020 −0.054, −0.013	−0.011 −0.052 −0.034	−2.742 −3.217 −3.268	0.008 0.002 0.002
NDI	Step 1 *Moderate at home*	0.258	0.004	0.002, 0.018	0.010	2.358	0.021
	Step 2 *Moderate at home* *Vigorous leisure*	0.252 −0.323	0.004 0.005	0.002, 0.018 −0.026, −0.006	0.010 −0.016	2.425 −3.117	0.018 0.003
	Step 3 *Moderate at home* *Vigorous leisure* *PPT*	0.264 −0.229 −0.373	0.004 0.005 0.076	0.003, 0.018 −0.021, −0.002 −0.440, −0.135	0.010 −0.011 −0.287	2.746 −2.304 −3.757	0.008 0.024 0.000
PPT	Step 1 *NDI*	−0.424	0.133	−0.814, −0.285	−0.550	−4.132	0.000
	Step 2 *NDI* *Cycling*	−0.409 0.224	0.130 0.085	−0.789, −0.271 0.021, 0.360	−0.530 0.190	−4.078 2.238	0.000 0.028
	Step 3 *NDI* *Cycling* *Gender*	−0.381 0.290 −0.506	0.108 0.071 2.531	−0.709, −0.280 0.105, 0.387 −20.383, −10.301	−0.494 0.246 −15.342	−4.593 3.472 −6.061	0.000 0.001 0.000
	Step 4 *NDI* *Cycling* *Gender* *Vigorous at home*	−0.425 0.295 −0.401 0.281	0.102 0.067 2.560 0.020	−0.756, −0.348 0.118, 0.383 −17.248, −7.049 0.027,0.108	−0.552 0.250 −12.148 0.067	−5.385 3.763 −4.746 3.335	0.000 0.000 0.000 0.001

Abbreviations: NDI: Neck Disability Index; PPT: Pain Pressure Threshold; VAS: Visual Analogic Scale. VAS: R^2^ adj. = 0.034 for step 1, R^2^ adj. = 0.106 for step 2, R^2^ adj. = 0.206 for step 3. NDI: R^2^ adj. = 0.055 for step 1, R^2^ adj. = 0.150 for step 2, R^2^ adj. = 0.273 for step 3. PPT: R^2^ adj. = 0.169 for step 1, R^2^ adj. = 0.210 for step 2, R^2^ adj. = 0.460 for step 3, R^2^ adj. = 0.524 for step 4.

**Table 6 biomedicines-10-02637-t006:** Summary of the Regression Analyses to determine salivary cortisol levels.

Targeted Variable	Predictor Outcome	*B*	SE B	95% CI	*β*	*t*	*p*-Value
**Salivary Cortisol**	Step 1 *Systolic Pressure*	−0.441	0.002	−0.012, −0.004	−0.008	−4.341	0.000
	Step 2 *Systolic pressure* *Vigorous at work*	−0.458 −0.299	0.002 0.001	−0.012, −0.005 −0.003, −0.001	−0.008 −0.002	−4.737 −3.091	0.000 0.003
	Step 3 *Systolic pressure* *Vigorous at work* *Moderate leisure*	−0.508 −0.141 −0.340	0.002 0.001 0.000	−0.013, −0.006 −0.002, 0.000 −0.001, 0.000	−0.009 −0.001 −0.001	−5.496 −1.364 −3.245	0.000 0.007 0.002
	Step 4 *Systolic pressure* *Vigorous at work* *Moderate leisure* *Vigorous at home*	−0.568 −0.561 −0.384 0.488	0.002 0.001 0.000 0.001	−0.014, −0.007 −0.005, −0.001 −0.001, 0.000 0.000, 0.003	−0.010 −0.003 −0.001 0.002	−6.078 −2.718 −3.708 2.330	0.000 0.008 0.000 0.022

Salivary cortisol: R^2^ adj. = 0.184 for step 1, R^2^ adj. = 0.265 for step 2, R^2^ adj. = 0.346 for step 3, R^2^ adj. = 0.382 for step 4.

## Data Availability

Not applicable.

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
