# Peer review of "Correlation among Routinary Physical Activity, Salivary Cortisol, and Chronic Neck Pain Severity in Office Workers: A Cross-Sectional Study"

_biomedicines, 2022, doi:10.3390/biomedicines10102637_

Round 1
Reviewer 1 Report
The reviewed manuscript is a research study on the correlation of physical activity, neck pain and salivary cortisol concentration in office workers. The topic is interesting because people working in a sitting position often report pain. The article is written in an interesting way, the methodical side does not raise my objections but there are some corrections which I recommend. Please find my comments below.
Abstract:
Lines 26-27: Please change this phrase into some brief conclusions of this study
Introduction:
Please describe why saliva is a good material for cortisol concentration determination.
Line 40: please describe cortisol secretion in more details – as well as HPA axis
Lines 64-69: please add information in which material cortisol was analyzed in mentioned studies
Materials and Methods:
Line 92: please clarify if this neck pain during 6 months had to be continuous or also shorter episodes were taken into consideration?
Line 95: why hormonal therapy was one of exclusion criteria? Please explain – maybe in Introduction?
Line 112: please add description how saliva samples were collected? How much saliva was in each sample? Were there any requirements for patients for preparation to sample collection?
Lines 113-119: In ELISA procedure there are at least 3 steps: incubation with enzyme conjugate (listed), incubation with TMB - not listed here and adding a TMB stop solution (listed). Please describe the protocol in more details.
Line 115: I guess it was an orbital shaker not a centrifuge? Please add device type and name of manufacturer
Line 116: please add the microplate reader type and the name of its manufacturer.
Lines 118-119: “considered a valid and reliable surrogate measure of the plasma levels of cortisol and the HPA axis activity [27-29].” – I recommend to move this part to Introduction (as I mentioned before)
Lines 119-120: This phrase needs to be rewritten – circadian rhythm refers to sample collection time, not an ELISA measurement.
Line 120: please add time frame for samples collection (were they collected in the moring? At what time?)
Line 133: for “valid” read: “validated”
Results:
Tables:
- please add description of abbreviations used in all tables (at the bottom of each table).
- please unify: “p” or “P”?
Discussion:
This part of the article should be extended – please see comments concerning each section.
Section 4.1
Please add more literature references concerning neck pain, p.ex.: https://pubmed.ncbi.nlm.nih.gov/21764519/ , https://journals.viamedica.pl/medical_research_journal/article/view/67853
Line 266: for “close” read “closely”
Lines 267-268: please describe the relation between the changes of listed parameters in cited article
Line 271: what did you mean by “pain cortisol disorders”?
Section 4.2
This part of Discussion is based on the same articles as Introduction – please modify Introduction to avoid repetitions. Consider adding more literature references to both parts of manuscript.
Section 4.3
This part of Discussion is based on the same articles as Introduction – please modify Introduction to avoid repetitions. Consider adding more literature references to both parts of manuscript.
Line 291: for “change” read “changes”
Lines 296-297: please extend the description of the results of cited article, p.ex.: “For example, yoga produces (…) strenghtening [17].” - and then mention the results of article [22]
Lines 299-300: This phrase is not related with the rest of the paragraph. Please add information concerning weight/BMI results in your study.
Please also add some literature references concerning physical activity level and cortisol concentration, p.ex.: https://pubmed.ncbi.nlm.nih.gov/18787373/ , https://www.frontiersin.org/articles/10.3389/fnbeh.2015.00013/full
Author Response
Response Letter manuscript biomedicines-1929984
Correlation among routinary physical activity, salivary cortisol and chronic neck pain severity in office workers: A cross-sectional study
We would like to thank the reviewers for their comments, which we believe have clarified many aspects of the manuscript. We have edited the text according to the suggestions from the reviewers. We have highlighted all changes in yellow throughout the manuscript. A point-by-point response is presented below.
Reviewer 1
Abstract:
Lines 26-27: Please change this phrase into some brief conclusions of this study
Response: We modified that sentence as follows: “Our results demonstrated the association between salivary cortisol concentration with moderate and vigorous physical activity, sitting time at work and PPTs. However, salivary cortisol was not associated with disability or pain intensity.”
Introduction:
Please describe why saliva is a good material for cortisol concentration determination.
Response: We agree this description was needed. However, we preferred reasoning the advantages of salivary cortisol measurement (compared with plasma measurements) in methods.
Line 40: please describe cortisol secretion in more details – as well as HPA axis
Response: We added the requested explanation in the second paragraph.
Lines 64-69: please add information in which material cortisol was analyzed in mentioned studies
Response: We clarified the methods. Three of the studies used salivary cortisol and one hair cortisol.
Materials and Methods:
Line 92: please clarify if this neck pain during 6 months had to be continuous or also shorter episodes were taken into consideration?
Response: We added the word continuous.
Line 95: why hormonal therapy was one of exclusion criteria? Please explain – maybe in Introduction?
Response: We decided to exclude participants under hormonal therapy since previous studies demonstrated whether contraceptive intake alters the circadian cortisol rhythm and attenuates salivary cortisol response as a response to stressful stimuli. A reference supporting this decision was added.
Line 112: please add description how saliva samples were collected? How much saliva was in each sample? Were there any requirements for patients for preparation to sample collection?
Response: We added the description:
“Patients were asked not to eat, drink, chew gums or brush their teeth 60 minutes prior to the sample collection and asked to rinse their mouth with cold water 5 minutes prior to the sample collection. A minimum of 1.0 mL of saliva was required for processing the sample. Samples with reddish color were dismissed due to possible blood contamination”.
Lines 113-119: In ELISA procedure there are at least 3 steps: incubation with enzyme conjugate (listed), incubation with TMB - not listed here and adding a TMB stop solution (listed). Please describe the protocol in more details.
Response: We revised the procedure and, as you pointed, some steps were missing. This is the complete procedure:
“All saliva samples were stored at -20ºC until analysis. To analyze the cortisol level in the saliva, four steps were followed. First, 50 μL of saliva was added to a tube using a pipette, and 100 μL of enzyme conjugate was added to each tube. Then, closed tubes were incubated at room temperature (18-25ºC) for 2h in a centrifuge at 400-600 rpm. Adhesive foils were removed, and the incubation solution was discarded. Plates were washed with 250 μL of diluted wash buffer and the excess solution was removed by tapping the inverted plate on a paper towel. Then, 100 μL of TMB substrate solution was pipetted and incubated 30 minutes at room temperature on an orbital shaker (again, 400-600 rpm). Afterward, 100 μL of TMB inhibitory solution was added (samples turned from blue to yellow), and results were measured using a photometer at 450 nm within 15 min after adding the last reagent.”
Line 115: I guess it was an orbital shaker not a centrifuge? Please add device type and name of manufacturer
Response: Thank you for pointing this misspelling. We revised and added the device information.
Line 116: please add the microplate reader type and the name of its manufacturer.
Response: We added this information as well.
Lines 118-119: “considered a valid and reliable surrogate measure of the plasma levels of cortisol and the HPA axis activity [27-29].” – I recommend to move this part to Introduction (as I mentioned before)
Response: We moved this part to the first paragraph of salivary cortisol assessment (Section 2.4.2.), immediately after the advantages of salivary measurements in comparison with other methods.
As we commented before, we believe that reporting all the diagnostic accuracy and instrumentation information would fit better in methods.
Lines 119-120: This phrase needs to be rewritten – circadian rhythm refers to sample collection time, not an ELISA measurement.
Response: We rephrased this sentence:
“Since salivary concentration is regulated by circadian rhythms, all sample collections were programmed at the same time (between 10:00 am and 10:30 am) for avoiding timing bias.”
Line 120: please add time frame for samples collection (were they collected in the moring? At what time?)
Response: This information was also specified.
Line 133: for “valid” read: “validated”
Response: We corrected.
Results:
- please add description of abbreviations used in all tables (at the bottom of each table).
Response: We apologize for omitting the abbreviatures in the tables. Tables 3, 4 and 5 have been revised
- please unify: “p” or “P”?
Response: We made consistent to “p”. In addition, we also unified numerical values from 0.XX to .XX.
Discussion:
This part of the article should be extended – please see comments concerning each section.
Section 4.1
Please add more literature references concerning neck pain, p.ex.: https://pubmed.ncbi.nlm.nih.gov/21764519/ , https://journals.viamedica.pl/medical_research_journal/article/view/67853
Response: Thank you for your recommendations. We addressed both references.
Line 266: for “close” read “closely”
Response: We modified.
Lines 267-268: please describe the relation between the changes of listed parameters in cited article
Response: This paragraph has been revised and we indicated the association’s directionality
Line 271: what did you mean by “pain cortisol disorders”?
Response: We deleted that sentence.
Section 4.2
This part of Discussion is based on the same articles as Introduction – please modify Introduction to avoid repetitions. Consider adding more literature references to both parts of manuscript.
Response: We used the studies referred in introduction to compare their findings with our results. We used new references (in addition to those used in introduction) for further discussion in this subheading.
Section 4.3
This part of Discussion is based on the same articles as Introduction – please modify Introduction to avoid repetitions. Consider adding more literature references to both parts of manuscript.
Response:
Line 291: for “change” read “changes”
Response: We corrected.
Lines 296-297: please extend the description of the results of cited article, p.ex.: “For example, yoga produces (…) strenghtening [17].” - and then mention the results of article [22]
Response: We provided a more detailed description:
“Cortisol secretion seems to oscillate depending on the time, intensity and specificity of the physical activity [17,22]. For instance, while 1-hour session of power or stretch yoga reduce similarly the salivary cortisol concentration (although power yoga was perceived more pleasurable and energizing in comparison with stretch yoga) [17], 3 tested core stabilization exercises (crook lying and rest, passive cycling in crook lying using automatic cycler and lumbar core stabilization exercise on a Pilates device) produced no changes in plasma cortisol concentration [22].”
Lines 299-300: This phrase is not related with the rest of the paragraph. Please add information concerning weight/BMI results in your study.
Response: Thank you, there was a missing text in this paragraph. We completed it as follows: “ Overweight and fatigue, which are closely associated with poor physical activity, [Jakicic JM, Davis KK. Obesity and physical activity. Psychiatr Clin North Am. 2011;34(4):829-840. doi:10.1016/j.psc.2011.08.009] are also reported to be positively associated with cortisol concentrations [18-20]. Since our ranges of body mass index and weight were limited, this may explain the inverse association found with salivary cortisol and therefore the associations found should be carefully interpreted”
Please also add some literature references concerning physical activity level and cortisol concentration, p.ex.: https://pubmed.ncbi.nlm.nih.gov/18787373/ , https://www.frontiersin.org/articles/10.3389/fnbeh.2015.00013/full.
Response: Thank you, we added these two references.
We hope that the current version of the paper can be finally accepted for publication in BIOMEDICINES
Sincerely yours,
The authors
Reviewer 2 Report
Comments:
1. Title: Is it ‘between’ or ‘among’?
2. Abstract: Line 14-15: the sentence seems incomplete. Please check it one more time.
3. Introduction, Lines 36-37: The sentence needs to be rewritten for grammatical clarity.
4. Lime 40-42: Please provide the supporting reference.
5. Similarly need supporting reference for the statement “while pain is a factor that could trigger the stress response by itself” as mentioned in Lines 43-44.
6. Line 67: What does it mean ‘medium term’?
7. Authors described the cortisol changes in the Introduction. Were all these studies focused on salivary cortisol?
8. Methods: Saliva collection details were missing. How was it collected? How much volume of saliva and what was the time of collection?
9. Results: Is it appropriate to use ‘patients’ to describe the participants? Please check and use properly.
10. PPT: Is it pain pressure or pressure pain threshold?
11. Table 2 and Table 3: Please describe the comparisons to indicate the significance (at the bottom of the tables).
12. What is the reason for mentioning only ‘systolic blood pressure’? Please discuss the results in the Discussion section.
13. How to rationale the study findings (salivary cortisol) with cortisol levels in the blood? Please discuss briefly.
14. Is it advised or not to perform physical activity at home? Or should it be tailored according to the physical status of an individual? Please discuss briefly.
Author Response
Response Letter manuscript biomedicines-1929984
Correlation among routinary physical activity, salivary cortisol and chronic neck pain severity in office workers: A cross-sectional study
We would like to thank the reviewers for their comments, which we believe have clarified many aspects of the manuscript. We have edited the text according to the suggestions from the reviewers. We have highlighted all changes in yellow throughout the manuscript. A point-by-point response is presented below.
Reviewer 2
- Title: Is it ‘between’ or ‘among’?
Response: Since correlations are not one-to-one, we also agree “among” is the correct term.
- Abstract: Line 14-15: the sentence seems incomplete. Please check it one more time.
Response: Thank you, we revised and completed.
- Introduction, Lines 36-37: The sentence needs to be rewritten for grammatical clarity.
Response: We revised: “In fact, up to the 50-70% of the entire population will experience (at least) one episode of neck pain clinically important throughout their life [2].”
- Lime 40-42: Please provide the supporting reference.
Response: We added this reference: Russell G, Lightman S. The human stress response. Nat Rev Endocrinol. 2019;15(9):525-534. doi:10.1038/s41574-019-0228-0
- Similarly need supporting reference for the statement “while pain is a factor that could trigger the stress response by itself” as mentioned in Lines 43-44.
Response: This reference was added: Tennant F. The physiologic effects of pain on the endocrine system. Pain Ther. 2013;2(2):75-86. doi:10.1007/s40122-013-0015-x
- Line 67: What does it mean ‘medium term’?
Response: Is referred to cortisol adaptations after 8 weeks.
“Regarding the medium term, previous research reported adaptations in cortisol secretion after an 8-week exercise routine”
- Authors described the cortisol changes in the Introduction. Were all these studies focused on salivary cortisol?
Response: We detailed those assessing salivary, plasma and hair cortisol.
- Methods: Saliva collection details were missing. How was it collected? How much volume of saliva and what was the time of collection?
Response: This information is now addressed.
“Since salivary concentration is regulated by circadian rhythms, all sample collections were programmed at the same time (between 10:00 am and 10:30 am) for avoiding timing bias. Patients were asked not to eat, drink, chew gums or brush their teeth 60 minutes prior to the sample collection and asked to rinse their mouth with cold water 5 minutes prior to the sample collection. A minimum of 1.0 mL of saliva was required for processing the sample. Samples with reddish color were dismissed due to possible blood contamination.
The instrument used for analyzing the saliva samples was a Cortisol kit RE52611 ELISA®. All saliva samples were stored at -20ºC until analysis. To analyze the cortisol level in the saliva, four steps were followed. First, 50 μL of saliva was added to a tube using a pipette, and 100 μL of enzyme conjugate was added to each tube. Then, closed tubes were incubated at room temperature (18-25ºC) for 2h in an orbital shaker (orbital shaker S-3.02 10L, ELMI SIA, Riga, Latvia) at 400-600 rpm. Adhesive foils were removed, and the incubation solution was discarded. Plates were washed with 250 μL of diluted wash buffer and the excess solution was removed by tapping the inverted plate on a paper towel. Then, 100 μL of TMB substrate solution was pipetted and incubated 30 minutes at room temperature on an orbital shaker (again, 400-600 rpm). Afterward, 100 μL of TMB inhibitory solution was added (samples turned from blue to yellow), and results were measured using the EASIA microplate reader (Medgneix, Fleurus, Belgium) at 450 nm reading absorbance within 15 min after adding the last reagent. ”
- Results: Is it appropriate to use ‘patients’ to describe the participants? Please check and use properly.
Response: “From 551 patients recorded…” Here we used the word “patients” since this is a clinical database. Those volunteers participating in the study are called now “participants”.
- PPT: Is it pain pressure or pressure pain threshold?
Response: We modified these misspellings to pressure pain threshold.
- Table 2 and Table 3: Please describe the comparisons to indicate the significance (at the bottom of the tables).
Response: We added a note for each table as requested.
- What is the reason for mentioning only ‘systolic blood pressure’? Please discuss the results in the Discussion section.
Response: We guess you refer this mention to those descriptions of Table 3 and 6. Since there was no association between cortisol and diastolic pressure (p>0.05), it could not enter the stepwise regression model (Table 6). This was explained in the statistical analysis processing.
- How to rationale the study findings (salivary cortisol) with cortisol levels in the blood? Please discuss briefly.
Response: We included this information in section 2.4.2.
“Measuring cortisol as a stress biomarker in saliva demonstrated to be a valid and convenient alternative to plasma cortisol measurement [Putignano P, Dubini A, Toja P, et al. Salivary cortisol measurement in normal-weight, obese and anorexic women: comparison with plasma cortisol. Eur J Endocrinol. 2001;145(2):165-171. doi:10.1530/eje.0.1450165] reflecting the HPA axis activity [27,28]. procedure. In addition, this method has several advantages in comparison with plasma measurements since samples can be collected during normal daily routines, stress-induced cortisol release responses are less likely to occur and fewer human resources are needed [Blair J, Adaway J, Keevil B, Ross R. Salivary cortisol and cortisone in the clinical setting. Curr Opin Endocrinol Diabetes Obes. 2017;24(3):161-168. doi:10.1097/MED.0000000000000328]. In addition, previous studies demonstrated its reliability [29].”
- Is it advised or not to perform physical activity at home? Or should it be tailored according to the physical status of an individual? Please discuss briefly.
Response: Physical activity is discussed separately for pain severity (Section 4.2.) and cortisol levels (Section 4.3.), comparing our results with other studies (observational studies, RCT, systematic reviews, meta-analyses…).
We hope that the current version of the paper can be finally accepted for publication in BIOMEDICINES
Sincerely yours,
The authors
Round 2
Reviewer 1 Report
Dear Authors,
I accept the manuscript in present form. Congratulations on your research!
Reviewer 2 Report
Appreciate authors for detailed responses.
Authors answered all the comments and manuscript has been revised accordingly. Now the manuscript can be accepted for publication.